# Assessment the Visual Clarity of the Projector in Classroom and Innovative Asymmetric Distribution LED Tube Applications

**Chun-Hsi Liu [1], Chun-Yu Hsiao [1],\*** , **Jyh-Cherng Gu [1], Kuan-Yi Liu [1], Chih-Hung Chang [2], Chen-En Lee [1] and Shu-Fen Yan [3]**

[1] Department of Electrical Engineering, National Taiwan University of Science and Technology (NTUST), Taipei City 106335, Taiwan; d10207103@mail.ntust.edu.tw (C.-H.L.); jcgu@mail.ntust.edu.tw (J.-C.G.); kim001533@yahoo.com (K.-Y.L.); M10907126@mail.ntust.edu.tw (C.-E.L.)

[2] General Manager of LIDlight INC., New Taipei City 235038, Taiwan; elvis@lidlight.com

[3] Department of Civil and Construction Engineering, National Taiwan University of Science and Technology (NTUST), Taipei City 106335, Taiwan; 10022682@mail.tycg.gov.tw

\* Correspondence: yuhsiao@mail.ntust.edu.tw; Tel.: +886-02-27333141 (ext. 6668)

**Abstract:** The paper aims to explore the relationship between the vertical plane luminance on projection screens and human visual clarity in the classroom or meeting room. While controlling the lighting environment conditions of the classroom to create different luminous distributions and luminance on the projection screen, a survey is conducted to understand students' visual experience about screen clarity during the field experiment. The luminance of each picture on the projection screen is measured under the specified lighting conditions of luminaires in the classroom, and the relationship is formulated between the average luminance on the projection screen and the visual satisfaction based on clarity of experience. This will be useful for further studying the acceptable threshold of luminance distribution in the classroom to provide a better visual clarity and lighting quality of projection screens while teaching. In this study, the measurement and performance evaluation on a projection screen were carried out at a classroom in the National Taiwan University of Science and Technology (NTUST). By using an image luminance meter and analyzing the research results, we propose an improvement strategy for asymmetric luminous distribution design of LED light tube and light switch control mechanism of luminaires to resolve the inadequate luminance of the vertical projection screen area to improve the lighting quality and visual clarity of the projection screen while teaching with the least cost.

**Keywords:** classroom lighting; luminance of vertical plane; projection screen area; screen clarity; average luminance

## 1. Introduction

Myopia and poor eyesight, an irreversible health defect, are the most serious physical defects of Asian students, especially in Taiwan and Mainland China. After the popularization of 3C electronic consumer products and the prevalence of mobile phones and tablets, excessive and improper behaviors and habits have led to a decrease in the age of myopia. The deterioration of myopia has accelerated, and the percentage of high myopia has also increased [1,2]. Han-Chih Cheng et al. [3] investigated the risk factors for myopia and their influence on the progression of myopia in schoolchildren in Taiwan.

According to the Myopia Survey conducted by Taiwan's Health Promotion Administration in 2017, the myopia ratio of first-year students in Taiwan's elementary school was 22%, and had risen to 66% drastically when it comes to sixth-grade. The myopia ratio of middle school students continued to increase to 77%, and by the time when students enrolled in high school, the myopia ratio is 85%. In addition, the number of myopia greater than 600 degrees accounted for one-fifth of the myopia population [4]. The percentage of

Taiwan students' poor eyesight rank third in the world, seriously impacting their health and educational experiences.

Using projectors to show slides or teaching with video have gradually become an indispensable and popular teaching mode both for teachers and students, since the teachers could save up a lot of time and provide much more information and sources to the students which help them to learn more efficiently [5]. However, the planning and designing of classroom lighting, the installation of luminaires and the grouping of power distribution control branches are mainly focused on energy-saving. Ingrid Heynderickx studied lighting influence on the visibility of projection screen and survey were carried out to collect desk illuminance and projection screen luminance case study data. The results indicated that the ambient illumination, the projection screen luminance, and the seating position all have a significant influence on the visibility of gratings on the projection [6]. Laura Bellia et al. [7] developed two methods to analyze comprehensive of the luminous environment, the former is based on the analysis of luminance maps obtained through the HDR imaging technique whereas the latter focuses on the evaluation of non-visual effects of light. Classroom lighting that takes into account the illuminance uniformity of the blackboard and students' desktops often causes excessive light intensity towards the projection screen during slideshow presentations, which seriously affect the, making it hard to correctly recognize the text on the screen. So far, the planning of the classroom lighting still lacks overall design guidelines and reference standards that engineers could follow when installing projectors and other related devices for teaching. Moreover, the study of projection screen clarity lacks quantitative evaluation and analysis [8]. When LED flat-panel luminaires, symmetric distribution shown in Figure 1, are introduced to indoor lighting project and replacing the existing fluorescent lamps luminaires, it has caused serious negative affect to the projection screen clarity, worsening the lighting quality of the vertical surface from the projection screen area of the classroom which affects the teaching quality and the willingness of students to learn. Vladimir Shalamanov et al. [9] examined the impact of LED light on workplaces in offices and classrooms. Although if the projection screen is replaced by a larger screen LCD monitor, the light pollution mentioned above could be resolved, but other issues will arise, for instance: the cost will be higher, and there is weight limitation for transportation and installation, etc. To conclude, using a projector while teaching is still a more suitable option compared to LED monitor when taking all the factors into account.

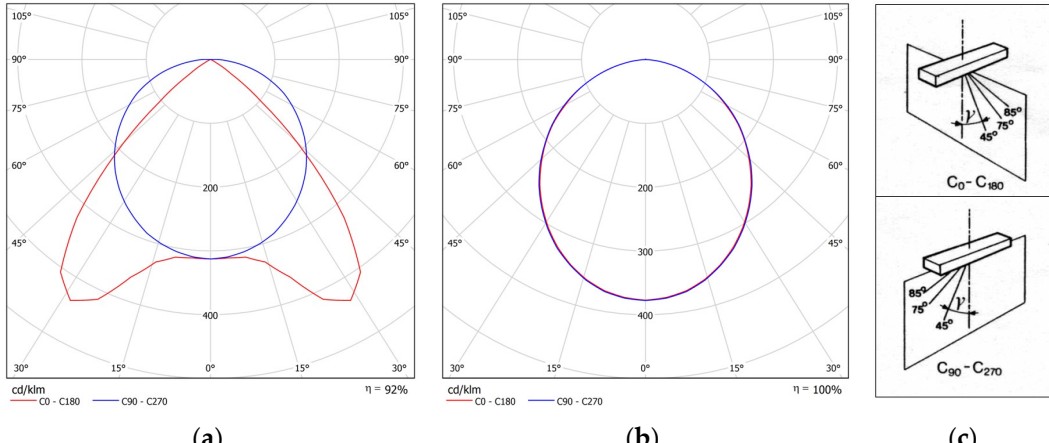

**(a)**              **(b)**              **(c)**

**Figure 1.** Light distribution of different kinds of luminaires: (**a**) conventional symmetric distribution, fluorescent or LED tubular luminaires, the red line refers to planes C0–C180, the blue line refers to planes C90–C270. Unit: cd/klm; (**b**) LED panel luminaires (right); (**c**) definition of distribution plane.

The clarity of the projection screen display is related to the luminance of the screen, as well as the luminance coming from the surrounding environment [10]. Antonio Peña-García et al. [11] proposed a quasi-Lambertian approach to real conditions in indoor

workplaces with a special aim in educative environments. The luminous flux (lumen) and light intensity projected by the projector are important factors related to the visual clarity of the projection screen image. If the luminous flux of the projector is insufficient, the lower the luminance value (cd/m$^2$) displayed on the projection screen, the lower the image luminance and the visual clarity. Moreover, due to the luminous influence of the luminaires coming from the desk area, the images and text on the screen will become much more difficult to recognize, since the resolution and clarity has worsened, as shown in Figure 2. The projectors equipped in classrooms generally output luminous flux of about 3000 lumens, which is sufficient for creating an acceptable and clear vision. Although the blackboard luminaires have been turned off to enhance the projection screen clarity when teaching, the luminaires around the classroom desk area are still turned on for students to take notes and writing purpose. Unfortunately, the light distributed by the symmetrical luminaires from the desk area illuminated the projection screen, which blurs the screen, making it very difficult for the students to recognize contexts projected [12,13]. Dave Coleman [14] presented both modeling and measurements of screen brightness and brightness uniformity. However, if all the luminaires above the desk areas are turned off, with only the projectors turned on, the image displayed on the screen will be clear. However, with high luminance contrast it will increase the risk of discomfort glare and is harmful to eyesight when undergoing long hours of teaching [15–17]. Piotr Pracki [18] evaluated impact of luminous intensity distribution of the direct lighting luminaires on ceiling and wall illumination, and discomfort glare. It is a consensus agreed upon that glare problem could be predicted and calculated by Unified Glare Rating (UGR), one of the main methodologies, which evaluated the luminance relationship between the watching target and surrounding environment. Jan Skoda et al. [19] described UGR measurement with luminance analyzer and discusses overall impact of background luminance on the calculation result. Jin Yang et al. [20] investigated the change of various indicators that reflect people's attention and visual fatigue under different indoor lighting conditions. Cheng Ruan et al. [21] propose a design strategy for the health light environment in classroom. Therefore, studying the luminance threshold required for the clarity of the projection screen and the light interference of the luminaires from the desk area is the first and foremost task when planning and designing a classroom lighting project. With these results, IoT technology can be applied in the future, and table-side sensing elements can be added to luminaires for intelligent dimming control [22–25].

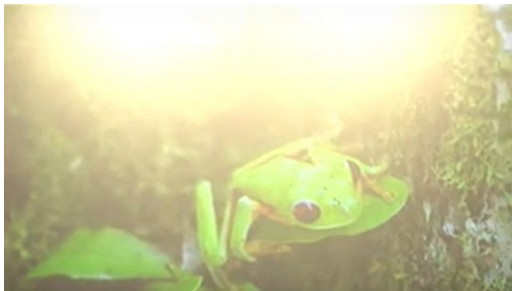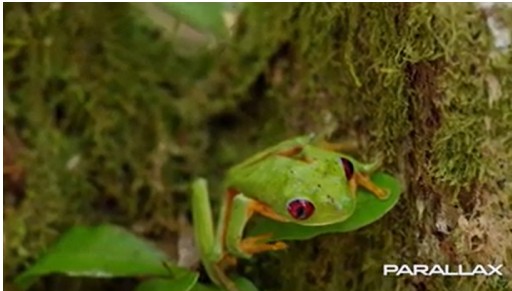

**Figure 2.** The lamp light affects the clarity of the projection screen. Reprinted with permission from ref. [26]. Accessed on 22 November 2021. Tsvetkova Darina.

In this paper, we propose an effective methodology for evaluating the visual clarity when teaching or presentation while using projectors in the classroom. For practical lighting improving engineering, we propose an innovative asymmetric luminous distribution design of LED light tube and light switch control mechanism of luminaires to improve visual clarity of the projection screen while maintaining the sufficient illuminance on the desk area for taking notes or writing purpose.

## 2. Materials and Methods

Regarding the relationship between the clarity of screen content, image and text, and luminance threshold, quantitative data survey and statistical analysis are required. Based on the objectives of this study, a classroom field experiment is carried out and students are participated to experience different projection screen clarity and luminance of the projection screen are measured during the whole experiment. Survey statistics data are collected from the students', which include their visual experience of the image on the projection screen under different light interference. Then, the relationship between the average luminance value of the projection screen and the students' perception of the clarity of the projected content under different controlled lighting conditions were summarized. The purpose is to formulate the functional relationship between the clarity of the projected image and the luminance of the image. Figure 3 is a flow chart of the assessment steps for this study, detailed experiments and on-site measurements are explained in the following sections.

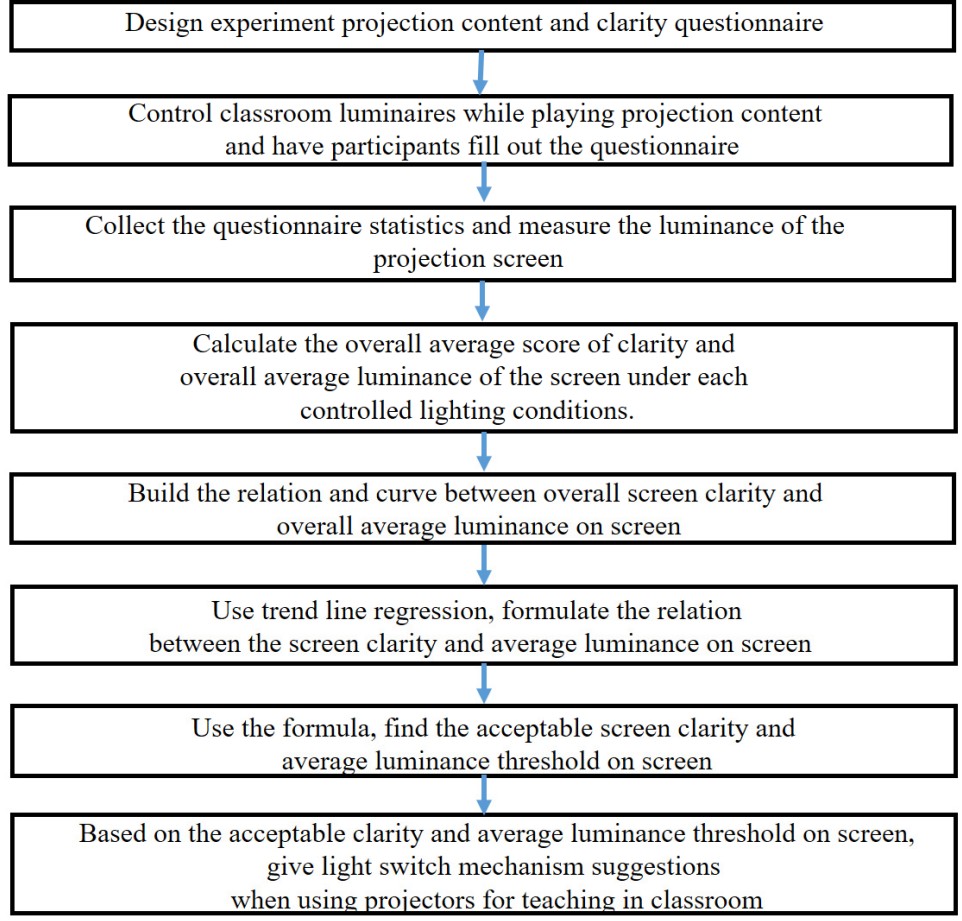

**Figure 3.** The flow chart for clarity study of the classroom projection screen.

### 2.1. The Selection of the Slideshow Content and Luminaires Switching Rules

In order to simulate the actual slideshow content used in teaching, the more slideshows selected, the more it can reflect the actual visual experience. The slideshow content should contain a wide range of colors, bright and colorful, high-definition pictures, dark colors and multi-colors, etc. However, if there are too many slides, the experiment time will last too long, which will cause visual fatigue of the participants and affect their subjective judgment regarding the clarity. In this study, we selected 6 pages of high-resolution slides, as shown in Figure 4, playing different visual scenes, showing both the daytime and nighttime picture for every scene; besides, you could also use the RGB primary colors with text to undergo the visual clarity experiments. Image contents should be selected carefully so

that the image luminance meter can still measure effective luminance data under different controlled lighting conditions making the experimental results much more accurate and of practical value.

Slideshow content

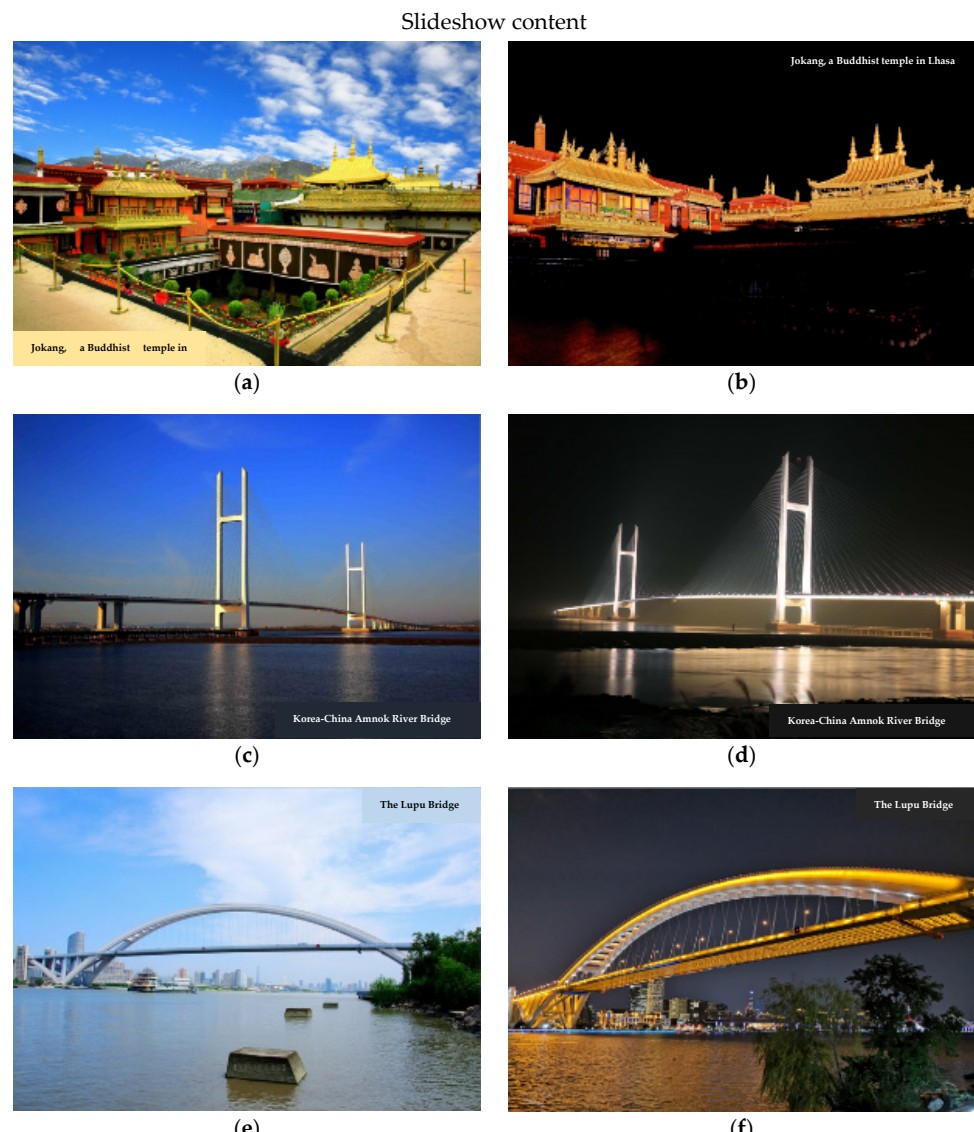

**Figure 4.** Examples of the slideshow contents. Reprinted with permission from refs. [27–32]. Accessed on 22 November 2021. (**a**) Temple landscape (day view), (**b**) temple landscape (night view), (**c**) cable-stayed bridge landscape (day view), (**d**) cable-stayed bridge landscape (night view), (**e**) arch bridge landscape (day view), (**f**) arch bridge landscape (night view).

The perception and acceptance of human vision for clarity varies from person to person. It would be meaningless if the grading system is consisted of too many scales. Therefore, in this paper, we designed a grading system for clarity perception survey: the blurriest will be given 2 points (20%); blurry ones will be 4 points (40%); normal ones will be 6 points (60%); clear ones will be 8 points (80%); and the clearest ones will be 10 points (100%) as shown in Figure 5.

The number and arrangement of luminaire in the classroom are related to the size of the classroom and the number of students. Generally, the luminaire of a 70 m$^2$ classroom have a row of blackboard luminaires and additional 9~12 luminaires lined up on the ceiling where the students are seated in their desk area. The luminaires in desk area were arranged by (3 × 3) or (3 × 4) usually, the blackboard luminaires will be turned off during the presentation of the slides, and the luminaire lined on the ceiling above the student are of

symmetrical distribution. According to the four switching modes of luminaires in desk area as shown in Figure 6, the light switch is used to adjust the different interference levels of the light to the slide content thus affecting the image clarity. While carrying out the experiment, when an assigned page of the slides is played, it will be influenced by the four different light control modes: with all of the luminaires in the classroom turned on, the luminaires in the first row of desk area turned off, the first and second rows turned off, and last, with all the luminaires in the classroom turned off. After the experiment, the participants can assess the clarity of the slideshow content accordingly. If the length of the classroom is long, we will only need to control the first four rows of luminaires, since the visual interference of the luminaires after the fifth row is very weak, so it is unnecessary to increase the luminaires control modes.

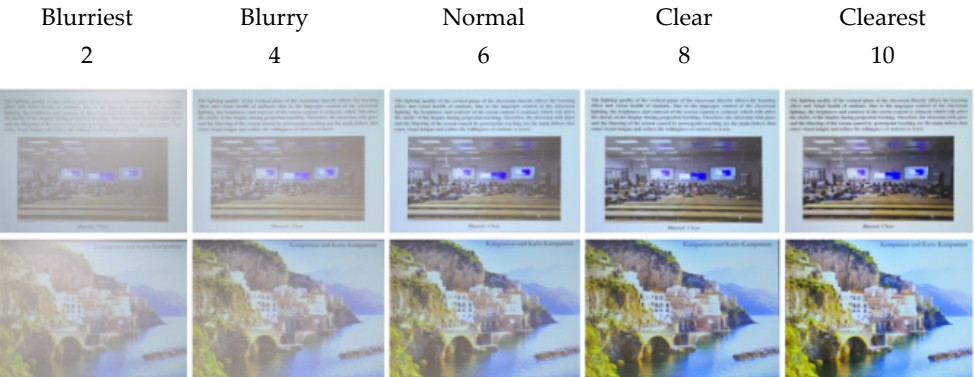

**Figure 5.** The scales of clarity.

Luminaires Control Modes

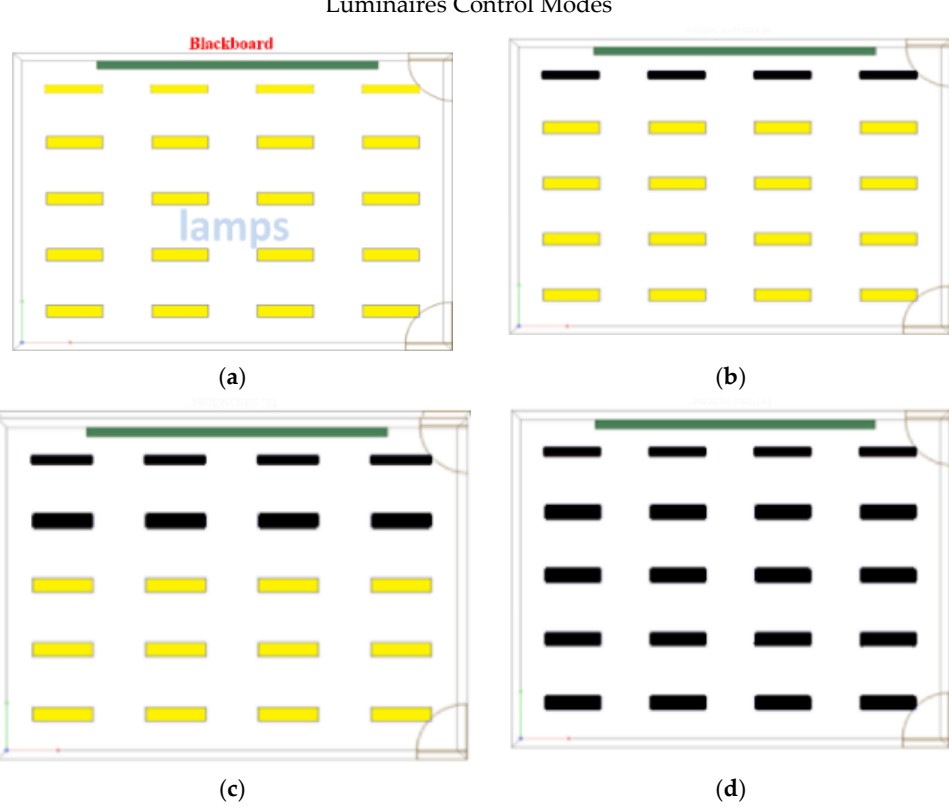

**Figure 6.** Schematic diagram for different luminaires control modes. Yellow means the light turns on, black means it turns off. (**a**) Level 1, (**b**) Level 2, (**c**) Level 3, (**d**) Level 4.

### 2.2. The Clarity Perception Experiment and Statistics of the Experimental Results

After the selection of slideshow content and the switching mode of the lighting control are determined, the feasibility of the preliminary experiment will be verified first. The selected content image will be projected in the classroom accordingly to different interference of light on the screen, students participating in the experiment will need to answer the questionnaire regarding the clarity of the projected content. The students' perception scores for different slide contents under different lighting control mode and lighting interference will be added up individually and divided by the number of participants. The scores represent the average score for clarity of each projection content under different lighting control mode. The college students that participated in the experiment is around 18 to 35 years old, which is the major era and group that has had much more experience with teachers using projectors to lecture. Therefore, the experiments results will have more reference value.

For each page of the slides, participants filled in the questionnaire and give scores according to their own subjective visual perception under different lighting control mode. Next, using Formula (1) to add up the average scores of all the projection content under specific lighting control conditions and divide it by the number of slides then we will obtain the total average score of the screen clarity under specific lighting control conditions when classroom teaching.

$$TAS_{(j)} = \frac{\sum_{m=1}^{k} AS_{(j)(m)}}{k} \tag{1}$$

$AS_{(j)}$: The average score of the perception for a certain slide $m$ under specific lighting control condition $j$
$k$: Total number of the slides in the experiment
$j$: Lighting control conditions
$TAS$: The total average screen clarity score under various lighting control conditions.

### 2.3. Image Luminance Meter and Luminance Measurment of the Projection Screen

The visual clarity of the scene can be evaluated by the level of luminance that enters the human eye. Therefore, an image luminance meter is used to simulate human eye to measure the luminance of the screen. The luminaires will be switched on and off one by one according to the sequences planned beforehand, and the image luminance meter is used to measure and calculate the visual luminance of each projection content under the control conditions of different lighting mode.

The image luminance meter is fixed around 110 cm height, which is equivalent to the eyes position of a student sitting down at the desk. The device is fixed at the center of the seating area to measure the luminance of the projection screen. The image luminance meter is LIDlight ILM-30, and the analysis software ILMA is set up with measuring point function (Figure 7) where 100 luminance measuring points are distributed on the projection content to calculate the luminance distribution, average luminance, highest luminance and luminance contrast of the projection content displayed on the screen, the data as shown in Table 1.

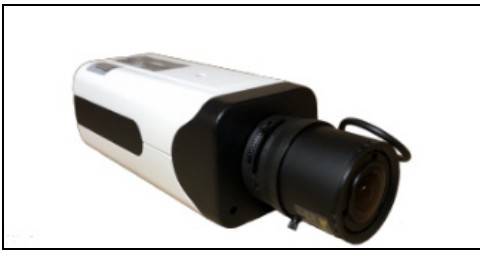
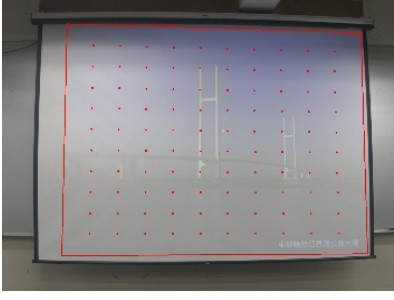

**Figure 7.** Image luminance meter and measuring point setting.

**Table 1.** The statistical results of the image luminance meter calculating the luminance level.

| Coordinate | Col 1 | Col 2 | Col 3 | Col 4 | Col 5 | Col 6 | Col 7 | Col 8 | Col 9 | Col 10 |
|---|---|---|---|---|---|---|---|---|---|---|
| Row 1 | 100.015 | 95.355 | 93.711 | 82.243 | 73.075 | 68.770 | 62.997 | 59.192 | 58.131 | 55.816 |
| Row 2 | 197.579 | 187.604 | 167.469 | 141.486 | 140.622 | 108.894 | 103.356 | 102.357 | 108.443 | 122.797 |
| Row 3 | 238.606 | 226.600 | 199.572 | 175.677 | 206.812 | 135.163 | 130.625 | 133.732 | 149.787 | 175.821 |
| Row 4 | 239.752 | 231.326 | 213.022 | 191.924 | 238.889 | 158.980 | 152.222 | 153.641 | 168.788 | 190.383 |
| Row 5 | 243.091 | 234.059 | 222.230 | 202.988 | 252.888 | 173.890 | 171.349 | 172.947 | 181.686 | 201.566 |
| Row 6 | 220.436 | 219.304 | 207.153 | 198.206 | 250.548 | 175.460 | 172.132 | 172.318 | 182.212 | 197.079 |
| Row 7 | 192.984 | 190.613 | 186.080 | 179.248 | 246.235 | 163.685 | 158.604 | 164.241 | 162.055 | 176.793 |
| Row 8 | 167.621 | 163.984 | 160.034 | 154.886 | 162.765 | 148.869 | 144.864 | 151.092 | 153.333 | 157.043 |
| Row 9 | 148.340 | 151.464 | 150.155 | 144.809 | 148.430 | 140.809 | 142.378 | 146.943 | 143.201 | 152.572 |
| Row 10 | 149.214 | 146.634 | 142.448 | 142.734 | 141.843 | 139.059 | 132.941 | 138.977 | 138.766 | 139.997 |

Luminance Average: 161.595
Luminance Maximum: 252.888
Luminance Medium: 158.980
Luminance Minimum: 55.816
Uniformity of Luminance U0: 0.345

Note. Unit: $cd/m^2$.

With the help of the image luminance meter, we can obtain the average luminance of the screen under different lighting control mode. We use Formula (2) to add up the average brightness of all the projection content under the specific lighting control condition and divide it by the number of slides, then the overall average luminance value of the projection screen when teaching is obtained.

$$TAL_{(j)} = \frac{\sum_{m=1}^{k} AL_{(j)(m)}}{k} \tag{2}$$

$AL_{(j)}$: The average luminance for a certain slide ($m$) under specific lighting control condition ($j$)
$k$: Total number of the slides in the experiment
$j$: Lighting control conditions
$TAL$: The overall average luminance of the screen under various lighting control conditions.

### 2.4. Establish a Quantitative Relationship between Screen Clarity and Visual Luminance

To obtain the overall average scores of the screen clarity of the classroom when teaching under specific lighting control conditions, we summed up the average scores of the clarity of all the projection content under the same lighting control conditions, and divided by the number of slides. Statistics data show that the overall average score of clarity under different lighting control conditions can reflect the impact of classroom lighting on the visual clarity of the slides.

In addition, under the same lighting control conditions, when the average luminance of all projection content measured by the image luminance meter is added up, and divided by the number of slides, then the overall average luminance of the screen can be obtained under a specific lighting control condition. The overall average luminance of the screen under different lighting control conditions can reflect the luminance level projected on the screen which is greatly impacted by the luminance of the classroom luminaires.

With this, we can establish the quantitative relationship between the overall average score of screen clarity and the overall average screen luminance under different lighting control modes in the classroom, as shown in Table 2, which shows the correlation relationship between visual clarity and screen luminance. As it is difficult for humans to distinguish 2% deviation from measurement, the overall average luminance value calculated was indicated by rounding with 2% deviation.

**Table 2.** Example of the overall average clarity score comparing to the overall average luminance value.

| Lighting Control Mode | Overall Average Luminance (cd/m²) | Overall Clarity Score |
|---|---|---|
| Level 1 | 180.72 (180 ± 2%) | 2.40 |
| Level 2 | 90.56 (90 ± 2%) | 4.61 |
| Level 3 | 52.14 (52 ± 2%) | 5.74 |
| Level 4 | 12.879 (12 ± 2%) | 9.41 |

### 2.5. Formulate the Relation between Screen Clarity and Average Luminance on Screen

First, with the data of Table 2, we use EXCEL to set the overall average luminance of the screen as the *x*-axis and the overall average score of the clarity as the *y*-axis to draw a relationship curve as shown in Figure 8. Second, by adding the approximate trend mode of the relationship curve to Figure 8 and through regression correction, the correlation between visual clarity and screen luminance is obtained as Formula (3), which can fully represent the relationship between the classroom projection screen clarity and average luminance. When using projection screen to teach, we came to realize that the visual clarity of the screen is affected by the interference of the lighting luminaires.

$$y = -2.61\ln(x) + 16.118 \tag{3}$$

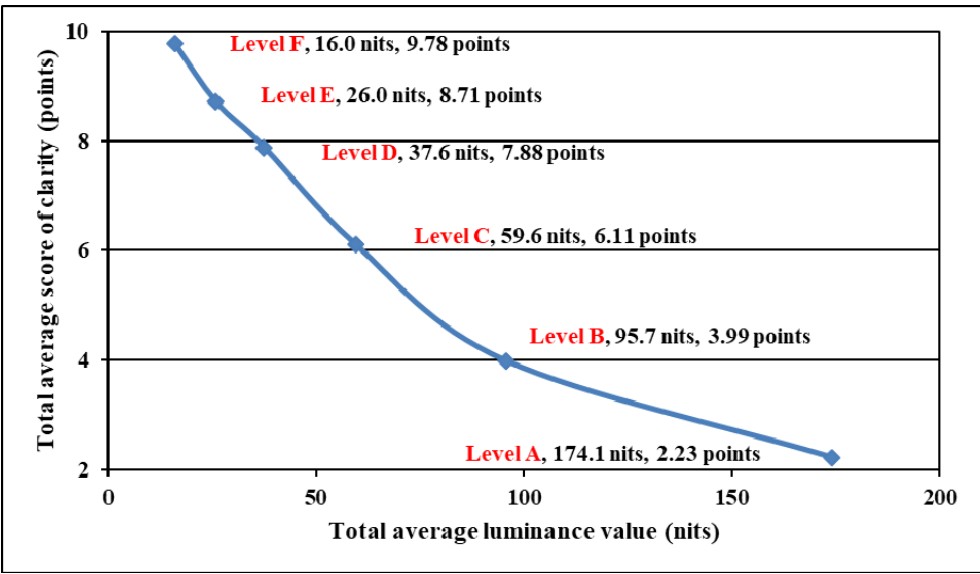

**Figure 8.** The relationship trend and formula between visual clarity and average luminance of the screen.

### 2.6. Visual Clarity Classification and the Corresponding Value of Average Luminance

After obtaining Formula (3) for visual clarity and average luminance by trend line regression method, we use the clarity scores: 2, 4, 6, 8, and 10 (as shown in Figure 5), to substitute y value of Formula (3), in order to obtain the corresponding average luminance value respectively as shown in Table 3. In the future, we can measure the average luminance of the screen by the image luminance meter, then calculate related the visual clarity, and to understand the impact of the luminaires on the screen, thus, developing effective improvement means, such as turning off the luminaires near the screen, or equipped the classroom with asymmetrical distribution luminaires for better visual clarity.

**Table 3.** Using clarity degrees to calculate average luminance.

| Human Eye Perception | y = −2.61ln(x) + 16.118 | |
| | y (Clarity) | x (Average Luminance, cd/m²) |
|---|---|---|
| The blurriest 20% | 2 | 223.45 |
| Blurry 40% | 4 | 103.85 |
| Normal 60% | 6 | 48.26 |
| Clear 80% | 8 | 22.42 |
| The clearest 100% | 10 | 10.42 |

*2.7. Discuss and Suggestions on the Lighting Control of Classroom Lighting Luminaires*

To evaluate the visual clarity of projection teaching, we can measure the average luminance on screen which corresponds to clarity level, and know whether the total average luminance under each lighting control conditions is clear enough for the students, to ensure effective teaching while using slideshows. The results can be used as a reference in practice of how to control the lighting mode when using the projection screen in teaching. The overall average luminance of level 1, where the classroom luminaires are fully turned on, is 180.72 cd/m² as shown in Table 4, and when referred to the x column of Table 5, the visual clarity of the human eye perception falls between blurry (103.85 cd/m²) and the blurriest (223.45 cd/m²), a bad and unacceptable projection quality. The average luminance of level 2, with the first row luminaires off, is 90.56 cd/m², and the clarity falls between normal (48.26 cd/m²) and blurry (103.85 cd/m²). The average luminance of the level 3, the first 2 row luminaires off is 52.14 cd/m², although the clarity falls between normal and blurry as is level 2, however it is closer to normal level (48.26 cd/m²), so level 3 is within the acceptable range of clarity for human eye. The total average luminance of the level 4, all luminaires off, is 12.879 cd/m², the clarity falls between clear (22.42 cd/m²) and the clearest (10.42 cd/m²), which is the best visual clarity. Therefore, when projection screen is used in the classroom, the best lighting control mode is level 4 where all the luminaires are turned off enabling the students to better (and clearly) recognize the projection content. However, the drawback of level 4 is that the luminance contrast ratio in the classroom will be too strong causing strong and serious contrast glare, which is unfavorable and unhealthy for students' visual experience. If we adapt level 3 and have the first two row luminaires closed, we can still maintain good projection screen clarity when teaching, reducing uncomfortable contrast glare, and provide students in the desk area enough lighting to take notes and writing therefore it is a better and much more suitable lighting control mode for classrooms.

**Table 4.** Luminaires specifications of classroom EE-406.

| Item | Specification | Distribution |
|---|---|---|
| Voltage (V) | 100~240 | |
| Frequency (Hz) | 60 | |
| Lumens (lm) | 2000 ± 10% |  |
| Efficacy (lm/W) | >100 | |
| Color temp. (K) | 5700 ± 300 | |
| Color rendering | >80 | |

A.    Lighting control method: The classroom is 10 m long and 11 m wide. The luminaires are arranged as $(5 \times 4)$. There are five rows of luminaires paralleled to the projection screen near the blackboard used for the switch control experiment.

    (a)    Level A: All of the luminaires are turned on.
    (b)    Level B: Turn off the first row luminaires.
    (c)    Level C: Turn off the first and second rows luminaires.
    (d)    Level D: Turn off the first, second and third rows luminaires.
    (e)    Level E: Turn off the first, second, third and fourth rows luminaires.
    (f)    Level F: All of the lights are turned off.

B.    Measure the luminance of the slideshow content: content pictures as shown in Table 1.

C.    Setting up the image luminance meter: 110 cm height from the ground.

**Table 5.** The overall average clarity score and the overall average luminance of the case study (classroom EE-406).

| Lighting Control | Overall Average Luminance (cd/m$^2$) | Overall Average Clarity Score (Points) |
|---|---|---|
| Level A | 174.117 | 2.23 |
| Level B | 95.666 | 3.99 |
| Level C | 59.552 | 6.11 |
| Level D | 37.577 | 7.88 |
| Level E | 25.951 | 8.71 |
| Level F | 15.985 | 9.78 |

## 3. Results

### 3.1. Case Study of Evaluating the Vertical Plane Clarity in the Projection Screen Area

We apply the measurement and evaluation methodology proposed above to measure the luminance of the projection screen area of the classroom, located at EE-406 classroom of the National Taiwan University of Science and Technology. The luminance data measured by the image luminance meter is used to evaluate the impact of the classroom's lighting on the image clarity. The study result can be further used to give suggestions for improving the projection screen image clarity in the future. The classroom uses 4-foot suspended symmetrical distribution LED luminaires as shown in Figure 9, and the detailed specifications of the luminaire are displayed in Table 4.

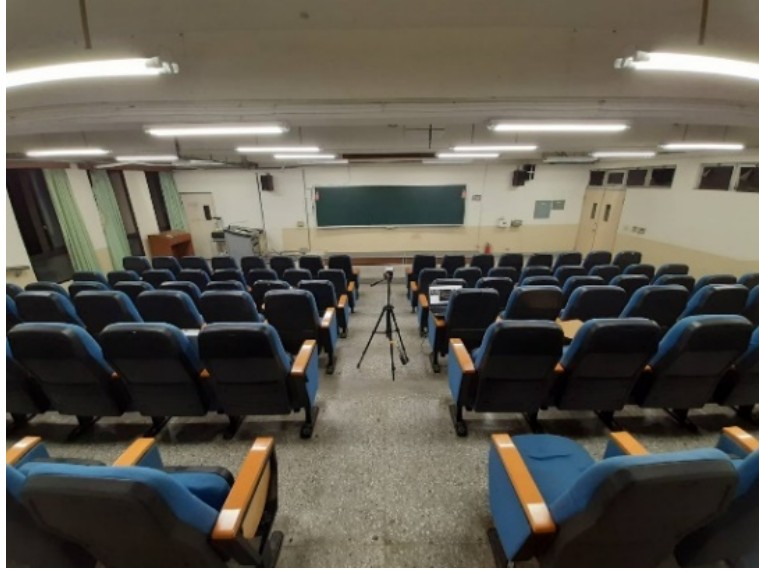

**Figure 9.** Application case (classroom EE-406) set up for clarity and luminance research.

In this study, a total of 60 students were asked to participate the clarity questionnaire survey. After the 60 participants marked their scores and summed up, the average clarity scores of the six slideshow contents within six different lighting control modes are shown in Figure 10, and the overall average scores of screen clarity under different lighting control modes are shown in Table 5.

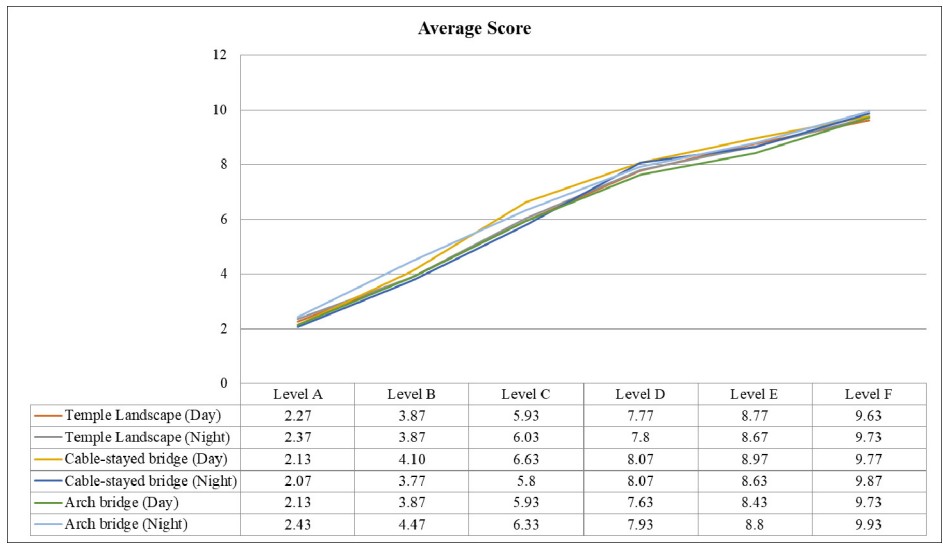

| | Level A | Level B | Level C | Level D | Level E | Level F |
|---|---|---|---|---|---|---|
| Temple Landscape (Day) | 2.27 | 3.87 | 5.93 | 7.77 | 8.77 | 9.63 |
| Temple Landscape (Night) | 2.37 | 3.87 | 6.03 | 7.8 | 8.67 | 9.73 |
| Cable-stayed bridge (Day) | 2.13 | 4.10 | 6.63 | 8.07 | 8.97 | 9.77 |
| Cable-stayed bridge (Night) | 2.07 | 3.77 | 5.8 | 8.07 | 8.63 | 9.87 |
| Arch bridge (Day) | 2.13 | 3.87 | 5.93 | 7.63 | 8.43 | 9.73 |
| Arch bridge (Night) | 2.43 | 4.47 | 6.33 | 7.93 | 8.8 | 9.93 |

**Figure 10.** Clarity questionnaire statistics of EE-406 (60 participants).

The average luminance of the six projection contents under different lighting control modes is measured by the image luminance meter. As shown in Figure 11, the luminance of the day scene picture is higher than the night scene picture under the same lighting control condition. The (luminance level) difference comes from the fact that the illuminance and luminance of the image during the daytime are inherently stronger than the nighttime illumination.

We use the average luminance data of each projection content measured under different lighting control modes as shown in Figure 11 and apply it to Formula (2) to obtain the overall average luminance of the screen when using slides to teach under various lighting control modes, as shown in the middle column of Table 5.

The overall average clarity score of each lighting control condition and the overall average luminance of the screen are listed in Table 5. The latter column is set as the *y*-axis, and the former column is set as the *x*-axis, one can draw the relationship curve as shown in Figure 12. After processing the trend line correction function, one gets Formula (4) which relates the overall average luminance of the screen and the degree of visual clarity for the classroom EE-406.

$$y = -3.3\ln(x) + 19.356 \tag{4}$$

It is evident in Figure 12 that the average screen luminance and the visual clarity score show an inverse proportion relationship. The higher the luminance on the screen, the lower the visual clarity. If the illuminance contributed by the luminaires projected on the screen is reduced, the luminance interference to the human eyes after reflection will also be reduced, which can effectively enhance the level of visual clarity. Physically, the human eye is not sensitive enough to clearly distinguish 10 lux of illuminance difference or 1 cd/m$^2$ of luminance difference, especially in the case of Figure 12 which the minimum luminance is 16 cd/m$^2$. Therefore, Formula (4) is sufficient to represent the correlation between the average luminance on screen and the level of clarity. Substituting the clarity grade score of the projection slides in Figure 5 into the y value of Formula (4), the average screen luminance corresponded is obtained as shown in Table 6. The comparison of lighting control mode with human eye clarity perceptions are listed as Table 7.

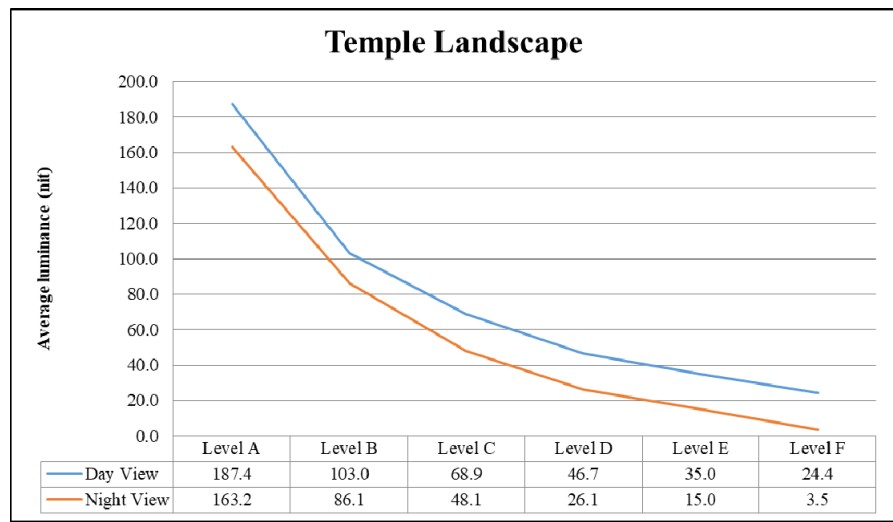

(**a**)

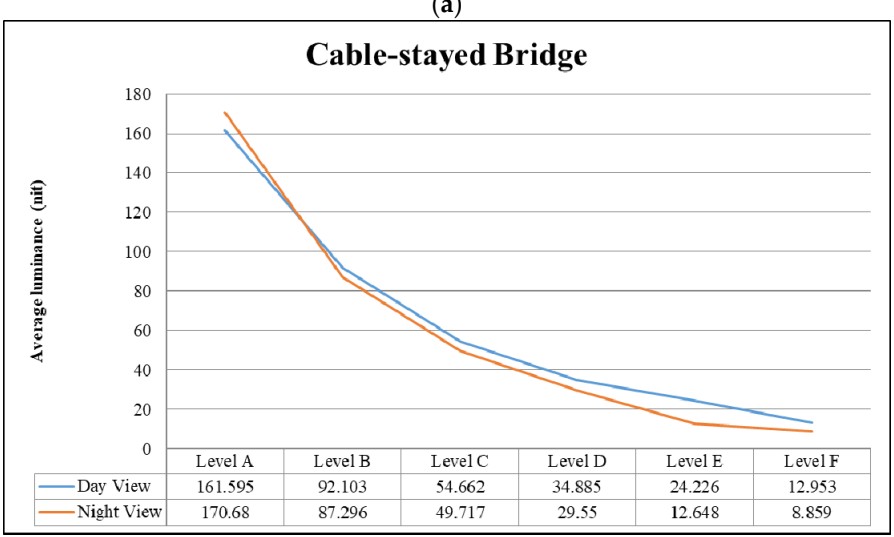

(**b**)

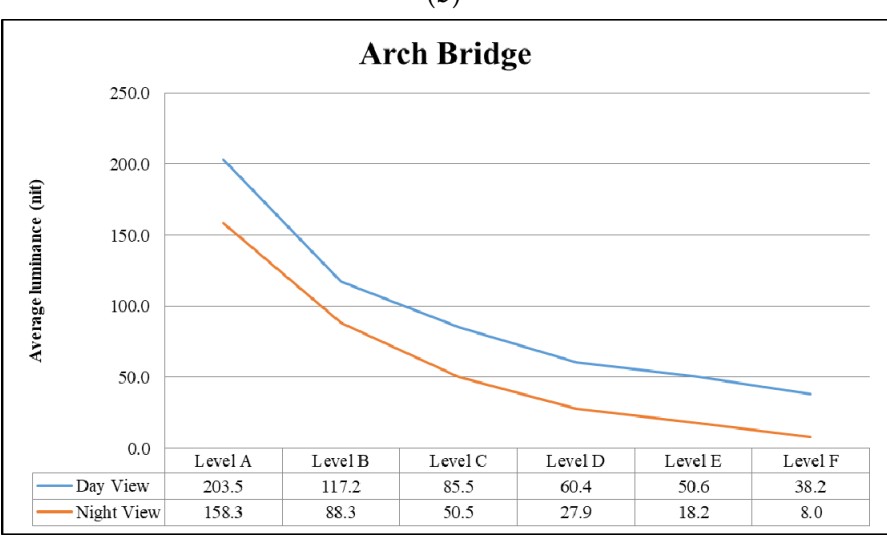

(**c**)

**Figure 11.** The average luminance of the: (**a**) temple landscape; (**b**) cable-stayed bridge; (**c**) arch bridge.

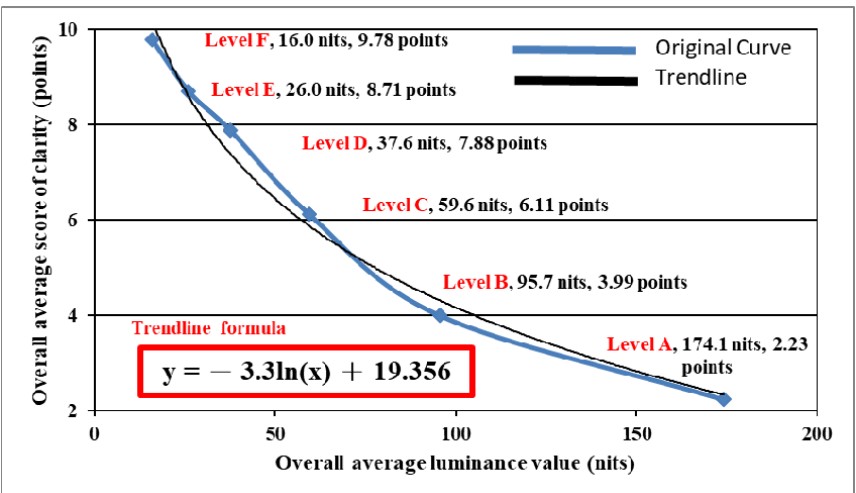

**Figure 12.** The trendline relates visual clarity and average luminance of the screen in case study.

**Table 6.** Clarity to average luminance threshold in case study.

| | $y = -3.3 \ln(x) + 19.356$ | |
|---|---|---|
| **Human Eye Perception** | **y (Clarity)** | **x (Average Luminance, cd/m$^2$)** |
| The blurriest 20% | 2 | 192.365 |
| Blurry 40% | 4 | 104.934 |
| Normal 60% | 6 | 57.241 |
| Clear 80% | 8 | 31.225 |
| The clearest 100% | 10 | 17.032 |

**Table 7.** Clarity evaluation of various lighting control in case study.

| Lighting Control | Total Average Luminance (cd/m$^2$) | Human Eye Perception |
|---|---|---|
| Level A | 174.117 (174 ± 2%) | Blurry/the blurriest |
| Level B | 95.666 (95 ± 2%) | Normal/blurry |
| Level C | 59.552 (59 ± 2%) | Normal/little blurry |
| Level D | 37.577 (37 ± 2%) | Normal/clear |
| Level E | 25.951 (25 ± 2%) | Clear and the clearest |
| Level F | 15.985 (15 ± 2%) | The clearest |

As Table 6 displays, in order to maintain better visual clarity on the projection screen during teaching or presentations, so as to provide students with better learning environment, it is advised to control the luminance level of projection screen below 50–60 cd/m$^2$.

Moreover, judging from Table 7, it is recommended to adopt level C and level D lighting control modes, so that students can recognize the projection content much more easily and clearly, meanwhile the luminaires in desk area can also provide basic lighting illuminance so that students are able to take notes, write or read books in the case study classroom EE-406 at the same time. When slideshow and presentation are in progress, it is suggested to avoid Level F with all the luminaires turned off which will cause serious contrast glare in a dark environment and damage the students' eyesight. The level E and level F modes are suitable mainly without writing and reading situations such as watching movies. Therefore, the best design of lighting configuration in the classroom should be planned with asymmetric distribution luminaires that can shield the light projected to the screen, which includes specially designed asymmetric distribution luminaires for desk area, and branches circuit arrangement to turn off the luminaires near projection screen in order to reduce the interference light effectively.

### 3.2. Innovative Asymmetric Distribution LED Tube and Lighting Configuration in the Classroom

In order to minimize the light influence from luminaires in desk area, and provide sufficient lighting illuminance on the desk at the same time, there should be at least 300 lux for reading and writing purpose. An innovative asymmetric distribution LED light tube is designed and implemented which can replace the conventional fluorescent lamps and LED light tubes directly, both symmetric distribution characteristics, and installed in existing symmetric distribution luminaires. The luminous distribution of the innovative LED tube is shown in Figure 13, which come from our design result by TracePro software and test report, and equipped in a normal classroom, the dimensions are 7.5 m (width) × 9.0 m (length) × 3.5 m (height). The luminous distribution of the luminaire is very similar to that of the LED light tube. We provide the luminous distribution of the LED tube just to present the asymmetric distribution feature.

The innovative LED tubes are installed in symmetric distribution luminaires in the normal classroom, and simulated by DIALux software, the results are given in Table 8. The result shows that the innovative LED tubes/luminaires provide satisfying illuminance that is greater than 300 lux whether the first row luminaires turned off or not. The illuminance overall the desk area, beyond the 2 m measurement point, is greater than 500 lux, sufficient for writing notes or reading books as shown in Figure 14. Figure 15 shows the 3D simulation graph of lighting environment. Meanwhile, if the first row luminaires be turned off, the luminance will as low as possible less than 10 cd/m$^2$ as shown in Figure 16, the clearest visual clarity with the least cost, we just replace the symmetric distribution LED tube with asymmetric distribution tube only!

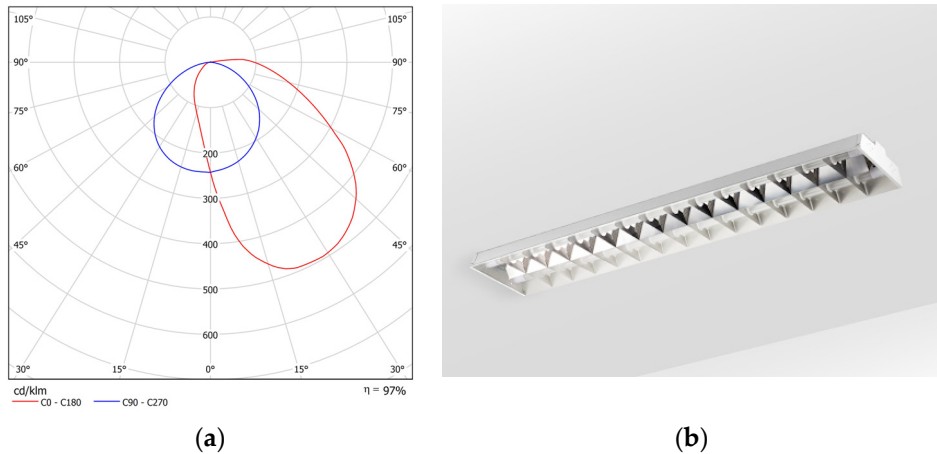

(**a**)  (**b**)

**Figure 13.** Asymmetric distribution of LED light tube (**a**) can be installed in conventional symmetric luminaires (**b**). The red line refers to planes C0–C180, blue line refers to planes C90–C270. Unit: cd/klm.

**Table 8.** Photometric data of the asymmetric distribution LED tube.

| Items | Level A: All Luminaires Turn On | Level B: 1st Row Luminaires Turn Off |
|---|---|---|
| Illuminance (lux), average | 489 | 445 |
| Illuminance (lux), (a) | 340 | 270 |
| Illuminance (lux), (b) | 583 | 556 |
| Luminance (cd/m$^2$), (c) | 17.25 | 13.15 |
| Luminance (cd/m$^2$), (d) | 9.39 | 7.11 |
| UGR vertical plane | 11.6 | ≤10 |
| UGR (d) | 11.0 | ≤10 |

Illuminance measured position: (a) 2 m away from screen; (b) center of the room; (c) screen wall (d); screen/blackboard area.

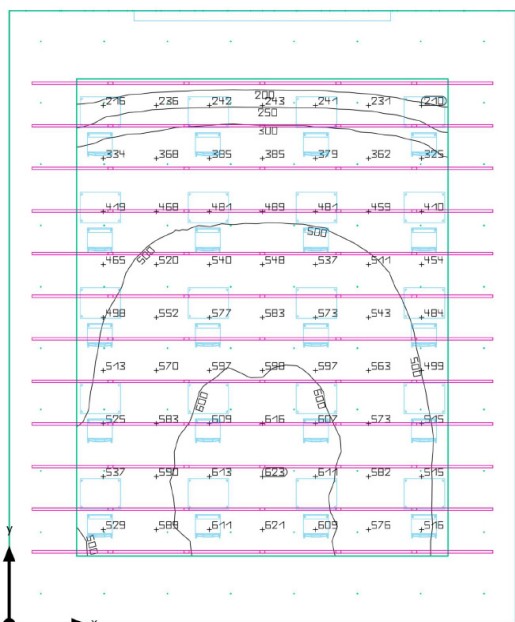

**Figure 14.** Illuminance distribution of the classroom by asymmetric distribution LED light tube.

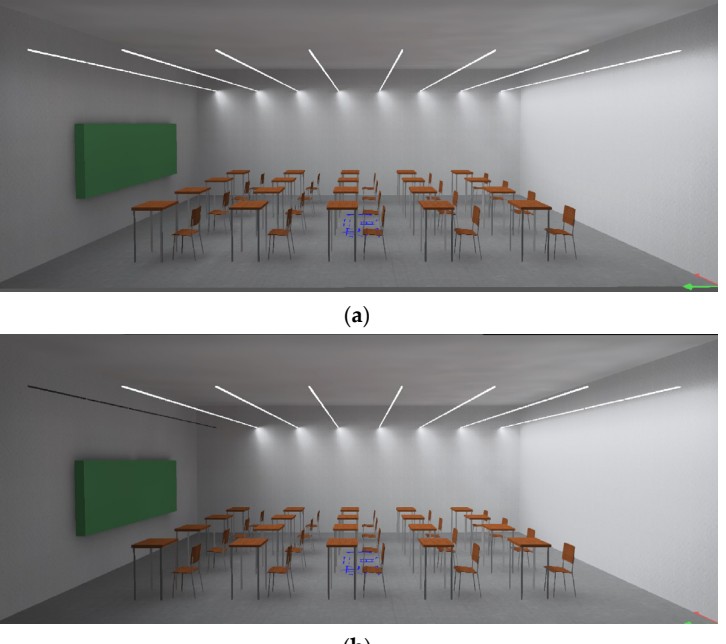

(**a**)

(**b**)

**Figure 15.** The 3D simulation graph of lighting environment. (**a**) all luminaires turn on. (**b**) 1st row luminaires turn off.

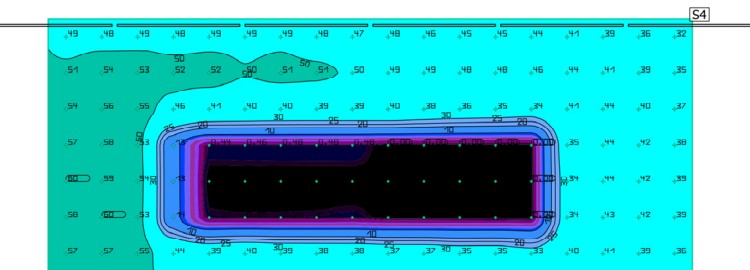

**Figure 16.** Illuminance distribution on the screen/blackboard by asymmetric distribution LED light tube.

## 4. Discussion

Bringing in teaching aids such as projectors and screens into classroom teaching is a major improvement for teaching methods, which could enhance teaching performances and reduce wasted time effectively; therefore, it comes without saying that the clarity of the screen and the comfort of the students' visual experience is very important in teaching. Theoretically, when the ambient illuminance is low, or in a pitch-black darkness room where luminaires are all turned off, the screen provides clearer image; but if the screen displayed a high luminance ratio when compared to its dark surrounding, it will cause glare which is harmful to eye health. Therefore, maintaining a low illumination of surrounding environment and allowing projection screen to present clear image quality is the key factors to the overall performance of teaching.

However, in the past, the lighting engineering in the classroom desk area usually uses symmetrical light distribution lamps and luminaires to achieve better illumination uniformity which leads to light interferences, caused by luminaires, on the projection screen. Therefore, contributing to lower contrast clarity of colors, text, and etc., on the screen. When LED panel lights are gradually becoming the main lighting source for classrooms, offices, and conference rooms, it is undeniable that the light interference to screen clarity will become more and more serious and affects the performances of teaching. However, for a long time, there has been little systematic research and quantitative evaluation on the effect of light distributions' affect to the lighting interference to projection screen in the desk area especially. Moreover, there is also a lack of a reasonable luminance value that takes into account the visual clarity of the screen and reasonable illuminance demands that allow students to take notes in class at the same time. Therefore, it is evident that most of the current classroom lighting quality is unable to meet the requirements for both teaching and learning.

This paper proposes a systematic survey and quantitative evaluation on projection screen clarity to find out the reasonable screen clarity and effective lighting switch control model in the classroom. In the study, in addition to providing a good lighting quality when using blackboard and projector screen in classroom teaching, we aim for better lighting design also. The lighting design could be applied to different places and conditions, for instance in conference rooms or even special needs for people of different age groups. In order to effectively improve the projection clarity and provide good illumination for students when taking notes in the classroom, this article proposes a novel LED asymmetric light distribution tube design, which could replace the existing symmetrical Light LED tube and fluorescent tube configuration without replacing the luminaires, which is fairly easy to implement. Hence, it can effectively reduce the light interference to the projection screen, and able to maintain a high-quality desktop uniform illumination; if students or teachers need to write on the blackboard, addition lights can be used to help reinforce the illuminance of the blackboard surface, however this is not within the scope of this article, nor does it affect the practicability of this article.

## 5. Conclusions

This paper explores the relationship between screen clarity and luminance of the lighting control in a normal classroom, equipped with symmetric luminaires, while using the projectors in teaching. The purpose is to undergo a quantitatively assessment about the impact on the screen clarity and the visual experience of students when the lighting is switched on and off. By investigating the subjective feelings of the experiment participants regarding the screen clarity of the selected slides under different lighting control mode, this research is able to establish a correlation trend and regression correlation formula between the luminance of the screen image and the clarity perception of the students. The given formula can be used to analyze and quantitatively assess what kind of lighting control mode that the teacher's should apply when using the projection screen to obtain better screen clarity and enhance students' learning effect.

It is proved from the two experiment cases in this research, that for better visual clarity on the projection screen during teaching or presentations, so as to provide students with better learning environment, it is advised to control the luminance level of projection screen below 40–50 cd/m$^2$. An innovative asymmetric luminous distribution LED light tube was designed and equipped in commercial symmetric luminaire, provided satisfying illuminance greater than 300 lux which is sufficient for writing notes or reading books, meanwhile, keeping the luminance as low as 10 cd/m$^2$, the clearest visual clarity.

The research proposes an assessment procedure and process of establishing regression statistics based on the level of visual clarity of the observer when perceiving the screen content and the screen luminance, which can quickly establish a set of lighting control suggestions that meet the visual needs of all age groups and various classroom teaching situations. This research could eventually be applied to the lighting environment quality assessment of all educational facilities.

**Author Contributions:** Conceptualization, C.-H.L.; methodology, C.-H.L. and K.-Y.L.; software, K.-Y.L., C.-H.C. and C.-E.L.; validation, C.-H.L., C.-Y.H., K.-Y.L., C.-H.C. and C.-E.L.; formal analysis, C.-Y.H. and K.-Y.L.; investigation, K.-Y.L., C.-H.C. and C.-E.L.; resources, C.-H.L., C.-Y.H. and S.-F.Y.; data curation, C.-H.L. and K.-Y.L.; writing—original draft preparation, C.-H.L., C.-Y.H., K.-Y.L. and C.-E.L.; writing—review and editing, C.-H.L., C.-Y.H., J.-C.G., K.-Y.L., C.-H.C. and S.-F.Y.; visualization, C.-Y.H., K.-Y.L., C.-H.C. and C.-E.L.; supervision, C.-H.L., C.-Y.H., J.-C.G. and S.-F.Y.; project administration, C.-H.L., C.-Y.H., C.-E.L. and S.-F.Y. All authors have read and agreed to the published version of the manuscript.

**Funding:** This research received no external funding.

**Institutional Review Board Statement:** Not applicable.

**Informed Consent Statement:** Not applicable.

**Data Availability Statement:** All data generated or analyzed to support the findings of the present study are included this article. The raw data can be obtained from the authors, upon reasonable request.

**Acknowledgments:** This research was funded by Ministry of Science and Technology, Taiwan (MOST 109–2221–E -011-051-MY2). Thanks to NTUST provide software and technical support.

**Conflicts of Interest:** The authors declare no conflict of interest.

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
