# Peer review of "Assessment the Visual Clarity of the Projector in Classroom and Innovative Asymmetric Distribution LED Tube Applications"

_applsci, doi:10.3390/app112311153_

Round 1

Reviewer 1 Report

I have included all comments and suggestions on manuscript I in the attached letter to the authors.

Reviewer 2 Report

I have analysed this manuscript which addresses the visual conditions in classrooms in Taiwan.

This paper is interesting in trying to balance screen illuminance with that provoked by the luminaires in the classroom environment.

The selection of the classroom is very arbitrary and its shape is quite basic it does not explore for instance, curved geometries.

Daylight seems to be totally out of the question in this paper. If there is in fact a decrease of the age in myopia, we should enforce more visual contact to the outside and different fenestration and skylight systems to better accommodate the eye in the classroom environment.

The images selected for checking visual response are also arbitrary.

That said, the methodology followed is correct and the calculations performed accurate.

 The authors need to clarify the concept of UGR. There is a minor disfunctions in Table 1.

The antecedents of the manuscript are sufficient. The examples selected seem slightly arbitrary as I said before, but they are not adequately implemented. On the other hand, several suggestions that  I have in mind for about the contents are:

More detailed 3D graphs should enhance the article.

The authors convincingly demonstrates the scope and possibilities of this research. The contribution is interesting for visual accuracy and environment.

The authors should perhaps simplify a bit the steps for their procedures and try to isolate their most outstanding findings.

The references are accurate and correct.

Summary of evaluation. Favourable because of the increasing potential of the demonstration. Accept with minor corrections that should be included in the next phase.

Reviewer 3 Report

This manuscript aims to explore the relationship between the vertical plane luminance on a projection screen and human visual clarity in the classroom or meeting room. While controlling the lighting environment conditions of the classroom to create different luminous asymmetric distributions for LED lamps and luminance on the projection screen. The idea is good and sounded; however, some issues should be addressed before publications according to the following comments:

1) The readability and presentation of this study should be further improved and simple to the reader. Please, correct the language problems, it is weak from the Grammarly and sequences of events, I catch 24 errors by using a personal program. The paper should be proofread very carefully by a native speaker or a proofreading agent.

2) The "Abstract" section should be more intensively and focused on the main idea directly and must contain the contribution of this manuscript. Also, please define the abbreviation NTUST in the "Abstract" section.

3) The "Introduction" section should be made much more impressive and focused on the main idea directly by highlighting your contributions. The novelty of this manuscript must be explained simply and clearly in points at the end of the introduction section. Note that, the introduction section should consist of three parts, i.e., general introduction to the topic, followed by literature survey, then the contribution clarifications.

4) In the introduction section, the literature survey should be enriched with the up-to-date references 2021 should be added and cited in the area of the latest trends in industry 4.0, smart building, and online wireless monitoring using the IoT and machine learning in lighting and other applications. E.g., Deep Learning-Based Industry 4.0 and Internet of Things towards Effective Energy Management for Smart Buildings & Reliable Industry 4.0 Based on Machine Learning and IoT for Analyzing, Monitoring, and Securing Smart Meters & Experimental Setup for Online Fault Diagnosis of Induction Machines via Promising IoT and Machine Learning & Towards Secured Online Monitoring for Digitalized GIS Against Cyber-Attacks Based on IoT and Machine Learning.

5) It is mandatory to check all the citing references of equations (1) and (2). In addition, check carefully all the abbreviation definitions and symbols in the whole manuscript. I catch some errors and the other symbols not defined.

6) The resolution and quality of many figures must be modified (e.g., asymmetric distribution of LED light in most figures); they should be presented as close to the camera-ready format. Further, please the authors should check carefully if there is any figure taken from another published paper it should be taken permission from the other publisher association, not only mention the citation reference. In additions photos were taken from Google Figure must add to the references section and cite them.

7) It will be helpful to the readers if various discussions about the methodological analysis and any other comparison previous studies are added.

8) The conclusion section should be rearranged, and numerical results should be added. Also, the authors may propose some interesting problems as future work in the conclusion.

Reviewer 4 Report

The authors report the assessment of the visual clarity of the projector in the classroom and innovative asymmetric distribution LED tube applications. The manuscript could be improved if address the following remarks:

  1. Please check the typos and grammar in the manuscript.
  2. In Figure 1(a), what do the red and blue curves represent?
  3. What are the units of the y axis in Figure 1?
  4. For Figure 5, the grading system for clarity perception was designed based on the naked eyes. How do you evaluate that this design was proper and optimal?
  5. How many participants were in this survey? Are they representative?
  6. In Table 1, how do you think the collected points were sufficient for the luminance measurement of the projection screen?
  7. In Figure 11, the curve legends are Chinese. Please replace them with English ones.

Round 2

Reviewer 3 Report

All my concerns are done, thanks.

Reviewer 4 Report

The authors have revised the manuscript carefully according to the requisitions and tried their best to resolve the problems in the comments. Therefore, I think that this revised manuscript should be accepted and published in this journal.